# Mining Heat-Resistant Key Genes of Peony Based on Weighted Gene Co-Expression Network Analysis

**DOI:** 10.3390/genes15030383

**Published:** 2024-03-21

**Authors:** Xingyu Yang, Yu Huang, Yiping Yao, Wenxuan Bu, Minhuan Zhang, Tangchun Zheng, Xiaoning Luo, Zheng Wang, Weiqun Lei, Jianing Tian, Lujie Chen, Liping Qin

**Affiliations:** 1College of Landscape Architecture, Central South University of Forestry and Technology, Changsha 410004, China; 20221100344@csuft.edu.cn (X.Y.); yyp18270327182@outlook.com (Y.Y.); 20221100345@csuft.edu.cn (W.B.); 20211200352@csuft.edu.cn (J.T.); 20211200306@csuft.edu.cn (L.C.); 2College of Art and Design, Nanning University, Nanning 530200, China; huangyu@nnxy.edu.cn (Y.H.); leiweiqun@unn.edu.cn (W.L.); qinliping@unn.edu.cn (L.Q.); 3School of Landscape Architecture, Beijing Forestry University, Beijing 100083, China; zhengtangchun@bjfu.edu.cn; 4College of Forestry, Henan Agricultural University, Zhengzhou 450046, China; wzhengt@163.com

**Keywords:** peony, transcriptome, weighted gene co-expression network analysis, heat resistance, enrichment analysis, core gene

## Abstract

The RNA-Seq and gene expression data of mature leaves under high temperature stress of *Paeonia suffruticosa* ‘Hu Hong’ were used to explore the key genes of heat tolerance of peony. The weighted gene co-expression network analysis (WGCNA) method was used to construct the network, and the main modules and core genes of co-expression were screened according to the results of gene expression and module function enrichment analysis. According to the correlation of gene expression, the network was divided into 19 modules. By analyzing the expression patterns of each module gene, Blue, Salmon and Yellow were identified as the key modules of peony heat response related functions. GO and KEGG functional enrichment analysis was performed on the genes in the three modules and a network diagram was constructed. Based on this, two key genes *PsWRKY53* (TRINITY_DN60998_c1_g2, TRINITY_DN71537_c0_g1) and *PsHsfB2b* (TRINITY_DN56794_c0_g1) were excavated, which may play a key role in the heat shock response of peony. The three co-expression modules and two key genes were helpful to further elucidate the heat resistance mechanism of *P. suffruticosa* ‘Hu Hong’.

## 1. Introduction

Peony (*P. suffruticosa Andr.*) is a deciduous shrub belonging to Paeoniaceae, Paeonia and *Sect. Moutan*. [1], which has the beautiful meaning of auspicious wealth [2] and magnificent [3,4]. The peony was first used to produce cortex moutan in Shaoyang, Hunan Province. It is one of the important medicinal cultivation areas of peony in China [5]. However, the planting effect of peony in southern China is not ideal. Due to the hot and humid climate, it is easy to cause stem and leaf damage, resulting in premature senescence of plants, affecting its root development, thus affecting its quality and yield [6]. The transcriptome of peony leaves under high temperature stress was studied for our research group, a large number of differentially expressed genes were obtained and enriched. The similarity between the biological replicates of the samples is high, and the differences between individuals are excluded. The data is convincing. However, due to the lack of complete genomic information, there are too many differential genes screened out. In order to solve this problem, we used the WGCNA method to modularly integrate them.

WGCNA is a system biology method used to describe the gene association patterns between different samples [7]. It can be used to identify highly co−variant gene sets (module), and the genes within the modules are highly positively correlated. Clustering is used to group genes with similar expression patterns in many samples [8]. These generated modules often represent a physiological process or a special phenotype. Functional enrichment analysis on the module was performed to see whether its functional characteristics are consistent with the research purpose. Then, the module and the sample were analyzed to find the module with a specific high expression of the sample. After that, further interaction information mining was performed on the selected modules to find the key genes needed.

Based on the transcriptome sequencing results, we used the WGCNA method to construct a weighted gene co-expression network and divide modules to explore the gene expression pattern of peony under high temperature stress and analyze the interaction information network of specific modules in order to explore the core genes of peony heat resistance and to lay the foundation for further study of the molecular mechanism of peony heat resistance.

## 2. Materials and Methods

### 2.1. Materials

The test material was *P. suffruticosa* ‘Hu Hong’, 2–3 years old. This variety belongs to the Jiangnan peony variety group, which is a peony variety with *Paeonia ostii* as the main wild ancestor [9]. It is widely used in the Jiangnan area and has good heat resistance.

The ‘Hu Hong’ peony seedlings used in this experiment were introduced from Ningguo, Anhui Province to the nursery base of Landscape Architecture College of Central South University of Forestry and Technology on 20 September 2020. The selected plants were potted (the diameter of the upper pot was 18.5 cm, the diameter of the bottom of the pot was 14 cm, and the height of the pot was 18.5 cm). The cultivation substrate was peat: vermiculite: river sand = 3:1:1 (*V:V:V*), and unified management in the same habitat. Because the peony has a good growth performance at about 20 °C, and the seed quality is high [10], reproductive development and nutrient exchange activities are good [11]. The temperature of 40 °C is the temperature at which peony leaves fall off at high temperature [12]. Therefore, it is set to be pre-cultured in May 2021 (about 20 °C, natural light, normal watering, a total of 15 days), and the plants are divided into two groups. The control sample (CK) and the sample treated with high temperature stress (HS) were placed in an artificial climate box at 20 °C and 40 °C, respectively. The humidity was 80%, and the light and dark conditions were 14 h (light intensity 12,000 lux)/10 h (light intensity 0 lux). The mature leaf samples of the CK group and the HS group were collected at five time points of 0 h, 2 h, 6 h, 12 h, and 24 h, and stored at −80 °C until RNA extraction.

### 2.2. Methods

#### 2.2.1. The Construction Process of WGCNA

Transcriptome raw data has been uploaded to NCBI, project number: PRJNA1079236, *P. suffruticosa* ‘Hu Hong’. The R package [8] in WGCNA is used to analyze and screen out the clustering modules with a high correlation degree. The Fragments Per Kilobase of exon model per Million mapped fragments (FRKM) of differentially expressed genes (DEGs) was analyzed by weight correlation analysis, and Pearson ‘s correlation coefficient (r) was calculated. In order to ensure that the gene connection conforms to the scale-free network distribution, we use the N-th power of the gene correlation coefficient, that is, the correlation coefficient weighting value. Secondly, the similarity between gene expression patterns is used for classification, which is used as an evaluation criterion for co-expression. By analyzing the different gene expression matrix in each sample, we can calculate the correlation between different genes, and derive the coefficient of gene difference, and module a heat map and system clustering tree intuitively to show the relationship between genes.

#### 2.2.2. Specific Module Screening

Functional enrichment analysis of genes in each module was performed using gene ontology (GO) and Kyoto encyclopedia of genes and genomes (KEGG) (FDR < 0.05). It is clear that the functional characteristics of each module are consistent with the research objectives. On this basis, through the correlation analysis of each module gene, the modules with high specific expression in the sample were screened out.

#### 2.2.3. Interaction Network Prediction of Protein−Transcription Factor and Key Gene Mining

Through visual analysis of the protein interaction network of key modules, the hub genes of each module and their strong clustering clusters were found. The model plant Arabidopsis thaliana was used as a reference species to compare the annotation results of the UniProt protein database with the specific genes in the module. The data of annotated genes in each module were analyzed by comparing the String database http://string-db.org (accessed on 1 May 2021), and PPI (protein−protein interaction) prediction network analysis was performed [13]. By determining the function of hub genes in the module and the categories enriched in GO function and KEGG pathway in the cluster, the core functions and genes of the module were clarified. Through the online String database, this study sorted out the transcription factors in the three key functional modules. Based on the reported and verified information about interaction transcription factors, the protein interaction network of core differentially expressed transcription factors in each module was constructed.

The protein interaction detection basis between modules was set to 0.95 (above the highest confidence level of 0.9 to remove low-related contiguity proteins), and the interaction detection basis between transcription factors was set to 0.7. Proteins with low correlation connectivity were eliminated, and the protein cluster analysis method was Markov Cluster Algorithm (MCL). The expansion parameter of MCL is 3, and the dotted line connects the protein clusters with interaction above 0.95 (higher than the highest confidence level 0.9) to obtain the interaction network diagram of protein and transcription factor.

## 3. Results

### 3.1. Construction and Module Division of Gene Co−Expression Network

The correlation is clustered using the interaction pattern between genes as a measure of co-expression. The gene expression matrix (FPKM) of DEGs in all samples was used as input data to calculate the adjacency between genes, so as to obtain the relationship between genes, deduce the dissimilarity coefficient between genes, and visualize the system clustering tree and module heat map between genes (Figure 1A). The branches in the hierarchical clustering tree correspond to modules, and the module membership of color coding is displayed in the color bar below and on the right of the tree. In the heat map, the high co-expression connectivity is represented by gradually saturated yellow and red. Modules correspond to highly interconnected gene blocks. Genes with high connectivity within the module are located at the top of the module branch, because they show the highest connectivity with the remaining genes in the module. Through mixed dynamic splicing, 19 gene modules were divided, containing a total of 15,599 different DEGs. In the correlation heat map of gene co-expression, the higher the correlation, the brighter the color. In the tree diagram, each branch represents a gene, and the ordinate is the clustering distance. The longer the branch, the greater the similarity of the expression profiles between different samples (processing time), that is, the clustering significance of the genes in the module is higher. It can be preliminarily inferred from the tree diagram that the gene clustering significance of the Yellow module is relatively the highest, while the gene clustering significance of the other modules is not strong. Correlation clustering is evident between modules. The heat map (Figure 1B) shows the correlation between modules. The modules under the same root are similar, and the color in the heat map is closer to red. Among them, the similarity between the Blue module and Tan module is relatively high, and the similarity between the Salmon module and Yellow module is relatively high. These four modules are relatively close in clustering. The Lightgreen module and Turquoise module, Pink module and Red module, and Brown module and Green module are relatively close in the clustering, and the similarity is relatively high.

### 3.2. Specific Module Screening

#### 3.2.1. Analysis of Module Characteristic Gene Expression Pattern

According to the relative expression of each module gene in each sample (FPKM), a histogram of the eigengene expression pattern of 19 modules was drawn (Figure 2). In the histogram, the column downward represents the relative down−regulated expression of the sample, that is, down-regulated expression, and the column upward represents the relative up−regulated expression, that is, up-regulated expression. We can analyze the dynamic expression trend of the module gene. In the diagram, we can initially identify the modules with significant differences in expression patterns between HS treatment and CK treatment. After excluding the modules with no obvious regular expression trend, it is not difficult to find that the Blue module, Salmon module and Yellow module showed significant up-regulation after heat treatment. The characteristic genes of the Brown module and Turquoise module showed a significant decrease in expression after high temperature treatment. The preliminary analysis of this type of expression pattern can provide a reference for subsequent specific module screening.

#### 3.2.2. GO and KEGG Enrichment Analysis of Module Characteristic Genes

After constructing the gene co-expression network module, the functional enrichment analysis of the genes in the module is the main basis for screening the key modules and their internal key genes. In the preliminary statistical results of the number of genes enriched by the GO and KEGG PATHWAY, the Turquoise module, Blue module, Brown module, and Yellow module were relatively enriched. In other modules, the Grey60 module, Lightcyan module, Lightgreen module, and Lightyellow module enriched relatively few genes (Figure 3).

GO functional enrichment analysis was performed on 19 modules, and the results enriched to various stress responses were selected from the enrichment results with FDR < 0.05 (Appendix A). When the enrichment results had more categories, the results of the top ten of FDR were selected for display. Among them, 19 modules contain functional enrichment of various stress responses. After preliminary screening, a total of 140 functional enrichment categories of various stress responses was obtained. After excluding biological stress sources (such as external biological stimuli such as various types of bacteria, viruses, microbial phytotoxins, chitin and other cellular components of organisms), ionizing radiation, mechanical stimulation and other stress responses, a total of 37 types of biological process enrichment items known to be related to plant high temperature stress were obtained (Appendix A). According to the enrichment results, 10 modules were involved in the ‘reaction to ethylene’ (GO: 0009737) and ‘reaction to abscisic acid’ (GO: 0009737), 9 modules were involved in the ‘reaction to salicylic acid’ (GO: 0009751), 8 modules were involved in the ‘reaction to jasmonic acid’ (GO: 0009753) and ‘reaction to karrikin’ (GO: 0080167). Preliminary analysis showed that with the extension of high temperature stress time, the genes involved in the regulation of endogenous hormone levels played an important role in the mechanism of plant response to heat. In addition, karrikin has been proved to be a small organic compound produced by plant combustion, and its enrichment in multiple modules is high. It is reasonable to speculate that the genes under this enrichment result are related to cell damage and heat damage of leaf scorch [14].

KEGG functional enrichment analysis was performed on 19 modules, and the results enriched to various stress responses were selected from the enrichment results with FDR < 0.05 (Appendix A). Among them, 19 modules, in addition to Cyan, Lightgreen, Magenta, Midnightblue, purple, Salmon, and the remaining 13 modules were significantly enriched in a total of 46 pathways (Appendix A). After preliminary screening, most of the seven module genes were significantly enriched in the photosynthesis pathway, and the Brown module had the highest enrichment significance (FDR = 1.99 × 10^−14^), indicating that the gene function of the Brown module may have the strongest correlation with photosynthesis regulation. The genes of the five modules were significantly enriched in the glyoxylic acid and dicarboxylic acid metabolic pathways, and the Turquoise module had the highest enrichment significance (FDR = 7.33 × 10^−11^), indicating that the gene function of the Turquoise module may have the strongest correlation between carbohydrate metabolism and oxidation reaction. The genes of the five modules were significantly enriched in the plant hormone signal transduction pathway, and the enrichment of the Brown module was the highest (FDR = 3.57 × 10^−12^), indicating that the gene function of the Brown module may have the strongest correlation with plant hormone signal transduction. The genes of the four modules were significantly enriched in the phenylpropanoid biosynthesis pathway, and the Turquoise module had the highest enrichment significance (FDR = 2.60 × 10^−6^), indicating that the gene function of the Turquoise module may play an important role in plant defense, structural support and growth and development. The genes of the four modules were significantly enriched in the photosynthesis−antenna protein pathway. The enrichment significance of the Brown module was the highest, and the FDR value was 2.4 × 10^−111^, indicating that the gene function of the Brown module may have the strongest correlation with the reception and transmission of light signals in photosynthesis.

#### 3.2.3. The Gene Expression of Each Module between Different Treatments

In order to further determine whether there were modules significantly related to high temperature treatment in the 19 modules, we tested the correlation between modular eigengenes (MEs) and the differences between treatments (Figure 4). The ordinate is 19 modules in the gene co-expression network. For each module, the heat map showed the correlation between the module feature vector gene (MEs) and the treatment group (10 treatments). The proportional bar on the right represents the possible correlation range from positive correlation (red) to negative correlation (blue). The Yellow module had the strongest extremely significant correlation with the phenotype in HS−2 treatment (r = 0.78, *p*-value = 3 × 10^−7^), and the Salmon module had the strongest extremely significant positive correlation with the phenotype in HS−6 treatment (r = 0.82, *p*-value = 3 × 10^−8^). In addition, the expression pattern of the Blue module in HS−12 and HS−24 treatment also had a significant positive correlation. However, the r values were relatively low, which were 0.6 and 0.54, respectively, and the *p*-value reached 5 × 10^−4^ and 0.002, respectively.

Through the preliminary judgment of the gene expression patterns of each module, the expression patterns of the five module genes of Blue, Salmon, Yellow, Brown and Turquoise were relatively regular. According to the enrichment significance in the heat-related GO functional classification of ‘response to heat’, ‘response to pressure’, and ‘response to injury’, the enrichment results of Brown, Cyan, Lightgreen, Lightyellow, Midnightblue, Salmon, Tan, and Yellow are close to the purpose of this study. Among them, the enrichment results of the Yellow module are highly significant and the enriched genes are abundant. According to the cluster analysis of the gene correlation of each module, the Yellow module and Salmon module are in the adjacent position of the same cluster, indicating that the gene expression patterns between them are highly similar. In addition, according to the correlation heat map between modules and treatments, the Yellow, Salmon and Blue modules with extremely significant and significant correlation with HS treatment were selected for analysis to narrow the screening range of key genes. In summary, the gene interaction network of the Blue, Brown and Yellow modules was selected for further analysis.

### 3.3. Prediction of Gene−Protein Interaction Network of Main Functional Modules and Mining of Key Genes

#### 3.3.1. Blue Module Protein Interaction Network

The Blue module contained 3414 genes, which were annotated to 2922 proteins in the UniProt database, forming a Protein−Protein Interaction Networks (PPI) relationship. The detection base of the interaction between proteins in the Blue module was set to 0.95 (higher than the highest confidence level 0.9). After removing proteins without high correlation connections, a total of 242 edges and 70 clustering interaction network diagrams of 156 gene nodes were obtained. The clusters of proteins with more than 0.95 interactions were connected by dashed lines (Figure 5). The top 30 clusters with the highest degree of interaction are shown in the figure. In the interaction network, it can be preliminarily seen that the degree of connection between clusters is not high, the functional consistency between clusters is not strong, and on the whole, it is more dispersed.

After screening, the key proteins predicted in the Blue module were AT3G04920 (14 interactions), EMB3137 (14 interactions), AT3G47370 (13 interactions), AT3G60770 (13 interactions) and AT1G23410 (12 interactions), which were blacked out in the protein interaction network diagram. Among them, AT3G04920, EMB3137, AT3G47370 and AT3G60770 belong to cluster 1 with the highest degree of connection, and the functional classification is the structural component of the ribosome and participates in mRNA binding. AT1G23410 is a lipoprotein S27 a, which belongs to the ubiquitin family protein. LHCA3 (light harvesting complex I chlorophyll a/b binding protein 3), LHCB4.1 (chlorophyll a/b binding protein CP29.1), PSAO (participating in the excitation energy balance between two photosystem I (PSI) and II (PSII)), PSAL (photosystem I reaction center subunit 11 OOV), the LHCB4.2 (chlorophyll a-b binding protein CP29.2) were hub proteins in the cluster 2. The GO function was enriched in chlorophyll binding and protein domain specific binding. It is mainly responsible for the function of capturing excitation energy in chloroplasts and balancing and transmitting it.

In summary, the hub genes involved in the Blue module may be closely related to the structural transformation of ribosomes and ribonucleic acid binding activities of peony under high temperature. Secondly, it is related to high temperature−induced cell heat adaptation, protein folding, and complex assembly. Finally, it is closely related to the activity of capturing excitation energy and balancing and transmitting it in chlorophyll binding.

#### 3.3.2. Brown Module Protein Interaction Network

The Brown module contains 113 genes and is annotated to 7541 proteins in the UniProt database. The annotation types are scattered and constitute a PPI relationship. Similarly, the detection basis of the interaction between proteins in the Brown module was set to 0.95 to remove the proteins without high correlation connections. A total of 276 edges and 96 clusters of 138 gene nodes were obtained. The clusters of proteins with interactions above 0.95 were connected by dotted lines (Figure 6). The top 30 clusters with the highest degree of interaction are shown in the figure.

After screening, the hub genes predicted in the Brown module were PSAK (22 interactions), PSAO (21 interactions), LHCA3 (20 interactions), LHCB4.1 (20 interactions), and LHCB5 (20 interactions). The five hub genes are blackened and circled in the protein interaction network diagram, all of which belong to cluster 1 with the highest degree of connectivity. Among them, PSAK is the photosystem i reaction center subunit psak, which is located in the chloroplast. PSAO and PSAK are both i reaction center subunits, which are involved in the excitation energy balance between the two photosystems I (PSI) and II (PSII). LHCA3 is chlorophyll a/b binding protein 3, and LHCB4.1 is chlorophyll a−b binding protein CP29.1. LHCB5 is a chlorophyll a-b binding protein CP26, chloroplast. The light-harvesting complex (LHC) acts as a photoreceptor, which captures the excitation energy and transmits it to the closely related photosystem.

#### 3.3.3. Yellow Module Protein Interaction Network

The 1309 PPI relationships are annotated by 1111 genes in the Yellow module. The detection basis of the interaction between proteins in the Yellow module was set to 0.95. After removing the proteins without high correlation connections, 252 edges of 87 nodes and 25 clusters were obtained. The interaction network diagram (Figure 7), and the clusters of proteins with interactions above 0.95 are connected by dashed lines. From the interaction network diagram, it can be seen that the degree of connection between clusters is high and the degree of clustering connection is good.

After screening, the hub proteins predicted in the Yellow module were AT4G16660 (11 interactions), CR88 (10 interactions), HSP90.1 (10 interactions), ATERDJ3B (6 interactions) and HSP101 (6 interactions), which were blacked out in the protein interaction network diagram. Among them, AT4G16660 is a HSP70 family protein, and its function is described as ATP binding. CR88 is a symbiotic ketone protein htpg family protein, which is a molecular chaperone required for chloroplast biogenesis. As a folding molecular chaperone (folding enzyme), HSP90.1 acts as a protein intermediate that maintains the function of the molecular chaperone (holdase) and can be stably unfolded, and quickly releases them in an active form after stress reduction. ATERDJ3B is a member 11 of the Dnaj homolog family b and belongs to the DNAJ heat shock family protein. HSP101 is an ATP-dependent CLP protease ATP−binding subunit CLPB, belonging to the chaperone protein CLPB1 (casein lyase/heat shock protein 100 (CLP/HSP100) family), which has been shown to be involved in the protein refolding of aggregates formed under heat stress. In addition, HSP70, which is also in the key position, is a possible medium for the transcription subunit 37c of RNA polymerase II. HSP70T−2 plays a role in stabilizing pre−existing proteins to prevent aggregation and mediates the folding of newly translated peptides in cytoplasmic gels and organelles.

Among them, HSP70 and HSP90 were mostly clustered in cluster 1, which was in the central position and the largest group. The rest include: Hop1 and Hop3 are stress-induced phosphoproteins 1 and 3 that mediate the binding of molecular chaperones HSP70 and HSP90. Hop3 has been shown to adapt to heat in plants such as Arabidopsis. ROF1 is a peptide−based proline isomerase FKBP62, which can accelerate protein folding. It catalyzes the cis−trans isomerization of proline−imide peptide bonds in oligopeptides (through similarity), and positively regulates heat-resistant common chaperones by interacting with HSP90.1 and increasing the accumulation of HSFA2−mediated small HSPs family chaperones. ROF2 plays an active role in tolerance to intracellular acid stress by interacting with FKBP62 and maintaining pH homeostasis [15]. MBF1C, a multi−protein bridging factor 1c, acts as a bridging factor between bZIP factor and TBP, and its expression is specifically increased in response to heat. In addition to the hub genes, *PsSgt1a* (TRINITY_DN63884_c0_g1, TRINITY_DN140134_c0_g1) has been shown to play an important role in the acquisition of heat tolerance in Arabidopsis [16]. There are various indications that the hub genes involved in the Yellow module are involved in or mediate protein folding with peony under high temperatures by up-regulating expression.

In addition, according to GO and KEGG PATHWAY enrichment results, the key genes in the Yellow module, *PsHSP70* (TRINITY_DN58827_c1_g1, TRINITY_DN58827_c1_g2, TRINITY_DN71223_c8_g1), was enriched in the GO category ‘biological process’ in ‘response to heat’, ‘response to hydrogen peroxide’, ‘response to high light intensity’, ‘response to cadmium ion’, ‘response to virus’, ‘response to temperature stimulation’, ‘response to bacteria’, and ‘protein ubiquitination’. At the same time, it was also enriched in ‘vacuole membrane’, ‘apoplast’, ‘cell wall’ of ‘cell component’, and ‘ubiquitin protein ligase binding’ of ‘molecular function’. In the KEGG PATHWAY enrichment results, these three *PsHSP70* were also enriched in the ‘MAPK signaling pathway’ (ko04010) and ‘endocytosis’ (ko04144) pathways. And verified by qRT-PCR, HSP70 was indeed highly expressed at 2h of high temperature stress. The difference is that TRINITY_DN58827_c1_g1 is enriched in the ‘chloroplast membrane’ of the ‘cell component’, while TRINITY_DN58827_c1_g2 and TRINITY_DN71223_c8_g1 are not enriched in this classification, which may indicate that the position of HSP70 may be closer to the related pathways of chloroplasts. In summary, *PsHSP70* is a key heat shock protein gene in the response of peony to high temperature stress, which is consistent with previous conclusions [17].

### 3.4. Prediction and Analysis of Transcription Factor Interaction Network of Main Modules

#### 3.4.1. Blue Module Transcription Factor Interaction Network

The detection basis of the interaction between transcription factors in the Blue module was set to 0.7 (high confidence). After removing the transcription factors without high correlation, a total of 49 edges and 17 clusters of 52 nodes were obtained (Figure 8).

According to the results, the most hub transcription factors involved in the interaction relationship are WRKY40 (6 pairs of interactions), GL3 (5 pairs of interactions), WRKY33 (5 pairs of interactions), IBH1 (4 pairs of interactions) and NF−YB3 (4 pairs of interactions). Among them, WRKY40 and WRKY33 are co-expressed in the same cluster, and co−expressed with ERF1, ERF11 and ERF6 transcription factors during stress, which are involved in ethylene signal transduction [18,19]. GL3 is a basic helix−loop−helix (bHLH) DNA binding superfamily protein, which is a transcriptional activator involved in the regulation of epidermal cell fate and negatively regulates stomatal formation. The co-expression network showed that GL3 may promote the formation of non−hair cells in N−position root epidermal cells together with MYB66 (WER) [20]. IBH1 is an atypical and possible non−DNA−binding bHLH transcription factor. As a transcriptional repressor, it responds to gibberellin (GA) and brassinosteroid (BR) signaling and has a negative effect on cell and organ elongation [21]. The Blue module co-expression network contains IBH1 and AT1G68920 (bHLH49). And IBH1 can form a heterodimer with bHLH49, thereby inhibiting the DNA binding of bHLH49.bHLH49 which is a transcriptional activator that binds to the G−box motif and inhibits HBI (a positive regulator of cell elongation), but there is insufficient evidence in the co-expression association between the two [22]. NF-YB3 is a nuclear transcription factor Y subunit B−3, which is a component of the NF-Y/HAP transcription factor complex, mainly involved in motif binding and stimulating transcription [23]. The members of this component are all CCAAT binding factor complexes in cluster 4, and the molecular function is enriched in the sequence-specific DNA binding of the cis-regulatory region of RNA polymerase II.

Overall, the main molecular functions of the Blue module transcription factors include DNA−binding transcriptional activator activity, RNA polymerase II cis−regulatory region sequence-specific DNA binding, and DNA−binding transcription factor activity. They play a key role in regulating carbohydrate utilization and inhibiting chloroplast elongation, and are involved in the active regulation of cutin, flavonol, and flavonoid biosynthesis processes.

#### 3.4.2. Brown Module Transcription Factor Interaction Network

The detection basis of the interaction between transcription factors in the Brown module was set to 0.7 (high confidence) to remove the transcription factors without high correlation. A total of 34 edges and 26 clustering interaction network diagrams of 38 nodes were obtained (Figure 9).

According to the results, the most hub transcription factors involved in the interaction relationship are WRKY40 (5 pairs of interactions), GL3 (4 pairs of interactions), NF-YB3 (4 pairs of interactions), NF−YC1 (4 pairs of interactions), and WRKY33 (4 pairs of interactions). It can be seen that the hub gene type of the Brown module is similar to that of the Blue module. The difference is that the Brown module hub transcription factor contains NF−YC1, but it is functionally the same as the nuclear transcription factor subunit. It has been confirmed that WRKY33 may be involved in the thermal response mechanism of Arabidopsis thaliana together with other WRKY family transcription factors [24].

Overall, the main molecular functions of brown module transcription factors include regulating RNA polymerase II general transcription initiation factor activity, DNA binding transcription activator activity and transcription co−activator activity, and they also play a key role in regulating carbohydrate utilization and inhibiting chloroplast elongation. They also participate in the biosynthesis of cutin and flavonols, and participate in the regulation of cell wall pectin metabolism. This indicates that the expression of key transcription factors in the Blue module may not only be involved in the regulation of the synthesis of compounds with stress resistance, but also may be related to the increase in cell membrane permeability in the peony heat stress response.

#### 3.4.3. Yellow Module Transcription Factor Interaction Network

The detection basis of the interaction between transcription factors in the Yellow module was set to 0.7 (high confidence). After removing the transcription factors without high correlation, a total of three edges and three clusters of six nodes were obtained (Figure 10). According to the results, the interaction of NF-YB3 and NF-YA3 in cluster 1 is mainly involved in motif binding and stimulating transcription. IAA26 and IAA16 in cluster 2 are mainly short-lived transcription factors in plant hormone signal transduction pathways. They are early auxin response genes that interact with ARFs at low auxin concentrations and play an inhibitory role. HsfB2b in cluster 3 is a winged helix DNA-binding transcription factor family protein that specifically binds to the transcription regulator of the DNA sequence 5 ‘−AGAAnnTTCT−3’, known as the heat shock promoter element (HSE). It is not only involved in the positive regulation of transcription by RNA polymerase ii, but also in the negative regulation of transcription and the biological process of the DNA template. When plants are induced by high temperature stress, the expression of the MBF1C gene is regulated by the HsfA1 factor [25]. In general, the main biological processes of the Yellow module transcription factors are involved in the positive regulation of meiotic cell cycle, the positive regulation of RNA polymerase ii promoter transcription on heat stress, and the response to chitin, which are mainly enriched in plant hormone signal transduction pathways.

According to the protein network prediction results of the Yellow module, the expression pattern of the transcription factors of the Yellow module was analyzed. There were 46 differentially expressed transcription factors (TFs) and 10 transcription regulators (TRs) in Yellow (Figure 11). Based on the weight results of the gene interaction relationship based on WGCNA, we construct the cytoscape network relationship according to the weight value of the relationship between genes and the relationship between the top 500 of the weighted weight values (Figure 12). By comparing these transcription factors with the key genes screened in the Yellow module co-expression network, only one transcription regulator Zinc finger protein CONSTANS-LIKE 13 (TRINITY_DN61637_c0_g1 ) was found. The gene was enriched into two sets of GO:’GO:0043565: molecular function−sequence specific DNA binding’ and ‘GO:0044212: molecular function−transcription regulatory region DNA binding’. However, it was not enriched in KEGG. In the gene co-expression network, Zinc finger protein CONSTANS-LIKE 13 is involved in three pairs of gene interaction. They were the relationship pairs with *PsHSP70* (TRINITY_DN58827_c1_g1) (weighted = 0.587), *PsClp_N* (TRINITY_DN65388_c4_g1, CLP amino terminal domain, virulence island component) (weighted = 0.584), and *PsHSP11/PsHSP12/PsSPP* (TRINITY_DN63985_c0_g2, chloroplast matrix processing peptidase) (weighted = 0.583).

## 4. Discussion

In this study, fresh leaves of *P. suffruticosa* ‘Hu Hong’ treated at room temperature and high temperature for different times were used as materials, and co-expression network analysis was carried out based on transcriptome data to mine key genes. It provides a reference for the subsequent research on the molecular mechanism of peony resistance to high temperature stress and the improvement of peony heat-resistant germplasm resources. The main conclusions are as follows:WGCNA is considered to be an efficient gene mining technology. It can specifically screen out the relevant genes that meet the requirements through the modular classification method, thereby obtaining a highly relevant co-expression module [26]. Zhang et al. [27] used the WGCNA method to divide and analyze the specific modules, and successfully excavated six candidate genes related to foxtail millet cold stress. Deng et al. [28] used WGCNA co-expression network analysis to predict 10 abiotic stress core genes in maize, and these genes may be the core genes of the abiotic stress module. Based on WGCNA analysis, Li et al. [29] excavated five key genes related to the anaerobic germination of rice as a second item.In the WGCNA network module, those genes that connect a large number of genes and have a high degree of connectivity are called hub genes. These hub genes play a crucial role in the network [30]. Through cluster analysis, it was determined that there was a high co-expression relationship between WRKY33, WRKY40 and WRKY53 transcription factors. Studies have shown that WRKY33 and WRKY40 can negatively regulate salicylic acid and jasmonic acid signaling pathways and biosynthesis [31]. However, the expression of WRKY33 and WRKY40 was not affected by high temperature stress, while the expression of PsWRKY53 was significantly different. It can be seen that PsWRKY53 may activate the key factors of the ethylene signaling pathway by co-expressing with PsERF6 and responding to the ethylene response element binding factor PsERF1A and the ethylene transcription factor PsERF11, thereby regulating the heat resistance defense response of peony. Using high-confidence screening of the Yel-low module, it was found that there was a strong co-expression relationship between AtHsfB2b and MBF1C proteins in cluster3. GO enrichment analysis showed that the genes of the Yellow module were highly enriched in the category of ‘ response to heat ‘. And it has been confirmed that the transcription factor can negatively regulate Arabidopsis thermomorphogenesis [32]. Therefore, it is speculated that PsHSFB2b may be a key factor for peony to resist heat stress.The key modules of the peony heat shock response were related to the functions of the Blue, Brown, and Yellow module. We predicted that *PsWRKY53* (TRINITY_DN60998_c1_g2, TRINITY_DN71537_c0_g1) and *PsHsfB2b* (TRINITY_DN56794_c0_g1) may play an important role in tree peony under high temperature stress. The WRKY53 transcription factor is mainly involved in plant endogenous hormone pathways. For example, the transcription factor was found to enhance plant stress resistance by regulating the gibberellin, jasmonic acid, and salicylic acid signaling pathways [33,34,35]. And WRKY53 has been reported to regulate plant growth and development, such as regulating the early and middle development of Arabidopsis leaves [36], leaf senescence [37,38] and so on. It can be seen that the WRKY53 transcription factor is widely studied in *Arabidopsis thaliana*, and the current research has not yet involved thermal response and response to ethylene signaling pathways. Therefore, whether *PsWRKY53* responds to ethylene signaling pathway and improves the heat resistance of peony needs further study. Previous studies have shown that HSFB2b plays an important role in plant response to stress conditions such as high temperature, salt, and drought stress [39]. As a transcriptional repressor, HSFB2b in Arabidopsis can regulate the thermomorphogenesis of Arabidopsis by inhibiting the expression of downstream heat-responsive genes at high temperature [40]. However, HSFB2b plays a positive regulatory role in pansy [41]. In this study, PsHSFB2b is located in the Yellow module, which is an expression pattern that first decreases and then increases. How to regulate the thermal response of red peony and its relationship with MBF1C protein needs further verification.

This study focused on the analysis of the heat tolerance related modules of peony and used the WGCNA method to analyze the correlation of the modules to mine two potential core genes related to high temperature stress. It laid a solid foundation for further exploring the relationship between core genes and the high temperature stress of peony and planned to verify the function of these genes in the future.

## Figures and Tables

**Figure 1 genes-15-00383-f001:**
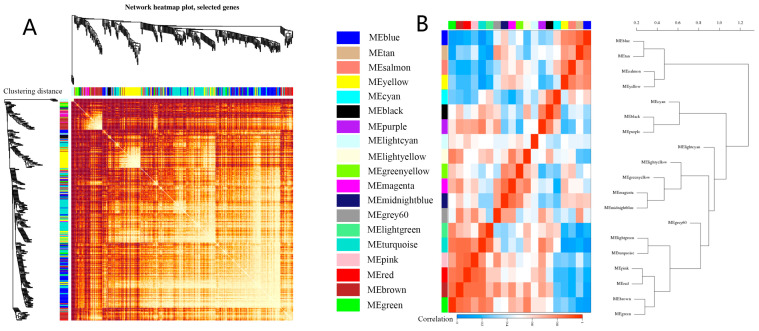
Clustering diagram of each module relationship. (**A**) Co−expression module clustering tree, (**B**) correlation clustering heat map between modules.

**Figure 2 genes-15-00383-f002:**
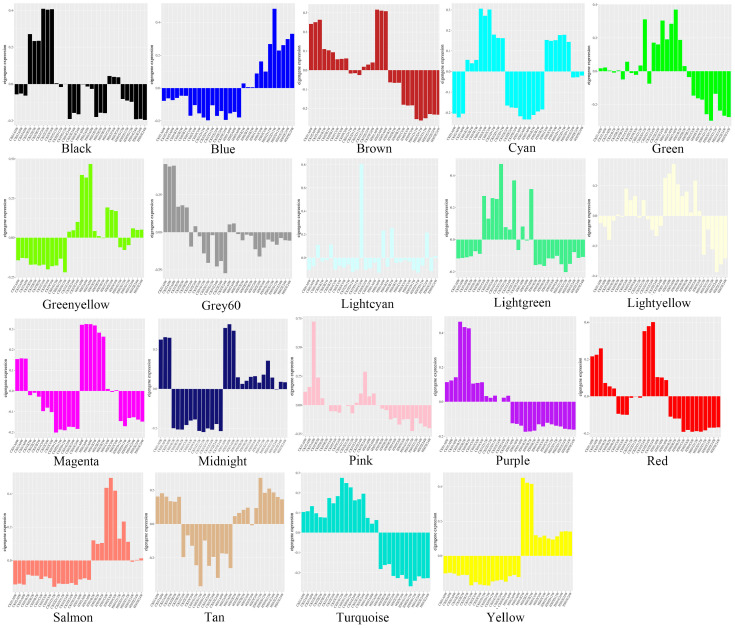
Expression patterns of eigengenes of 19 modules.

**Figure 3 genes-15-00383-f003:**
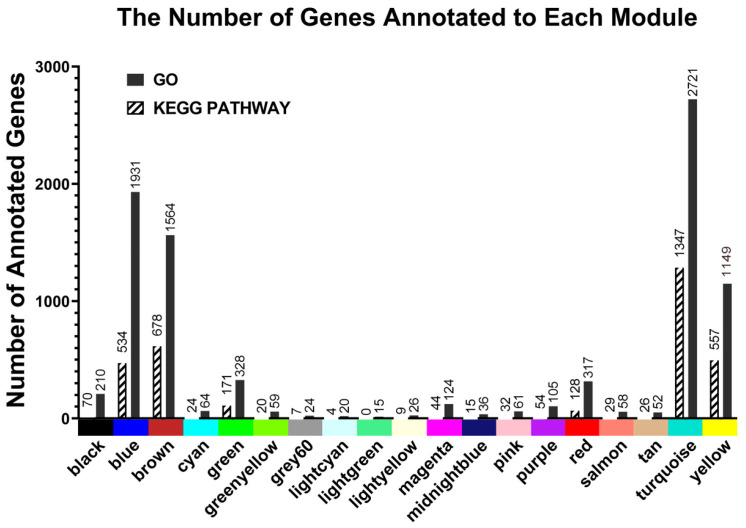
Enrichment results of eigengenes in GO and KEGG PATHWAY.

**Figure 4 genes-15-00383-f004:**
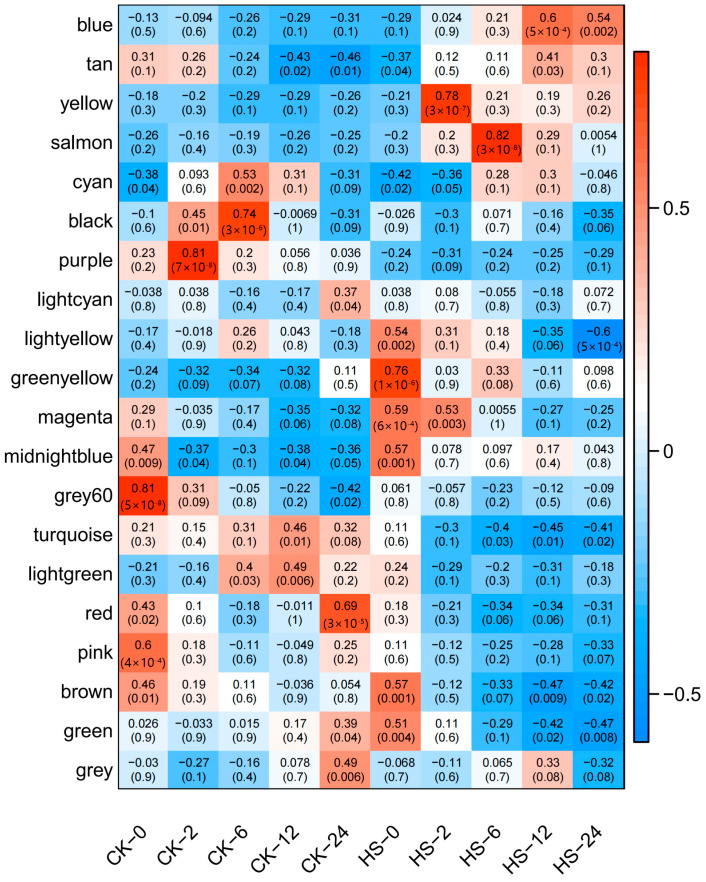
Heat map of the correlation between modules and traits. The cell number represents the progressive *p*-value (corPvalueStudent) and correlation coefficient of the relevant research. The module screening threshold corresponding to significant correlation was r^2^ > 0.6.

**Figure 5 genes-15-00383-f005:**
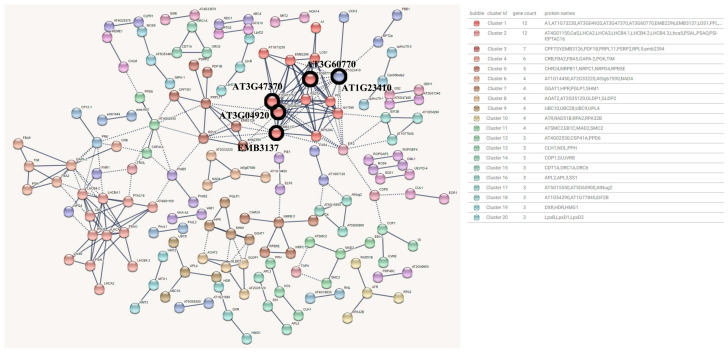
Blue module protein interaction network prediction diagram.

**Figure 6 genes-15-00383-f006:**
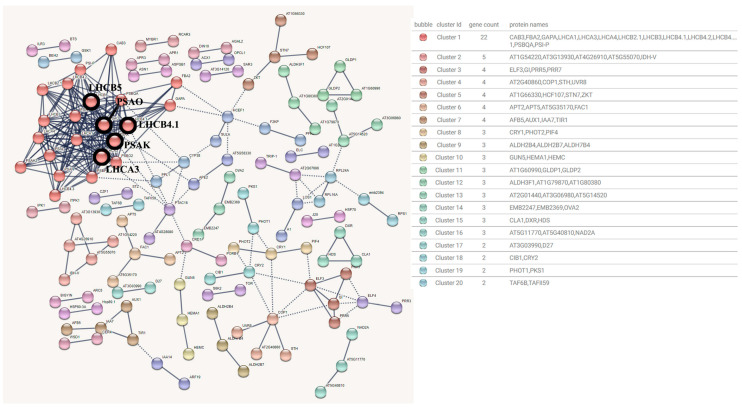
Brown module protein interaction network prediction diagram.

**Figure 7 genes-15-00383-f007:**
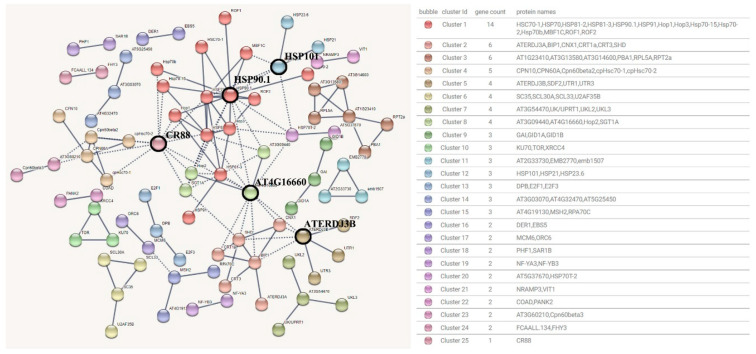
Yellow module protein interaction network prediction diagram.

**Figure 8 genes-15-00383-f008:**
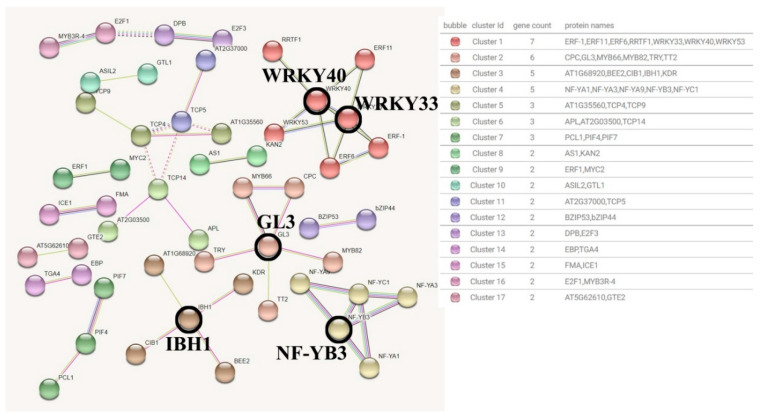
Blue module transcription factor interaction network prediction map.

**Figure 9 genes-15-00383-f009:**
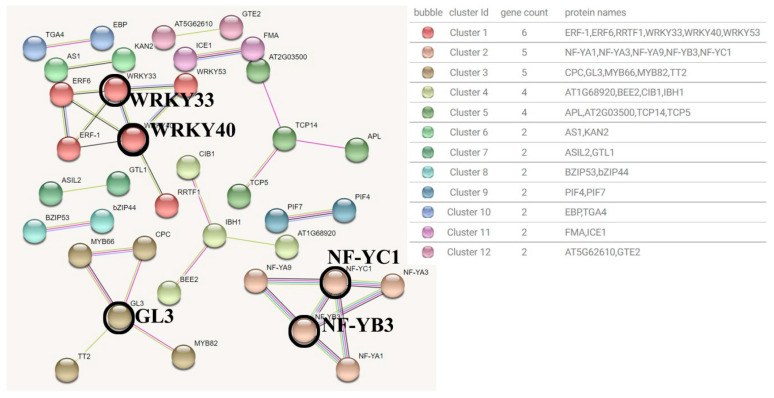
Brown module Transcription factor interaction network prediction map.

**Figure 10 genes-15-00383-f010:**
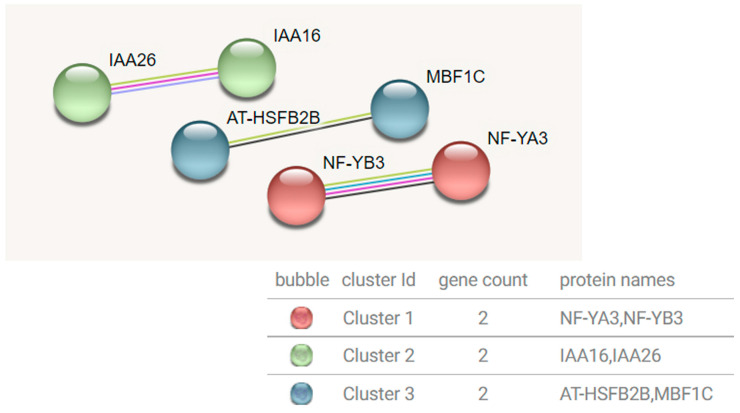
Yellow module transcription factor interaction network prediction map.

**Figure 11 genes-15-00383-f011:**
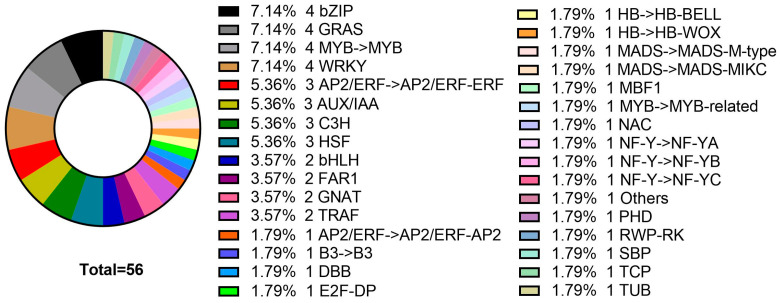
Statistics of transcription factor genes in Yellow module.

**Figure 12 genes-15-00383-f012:**
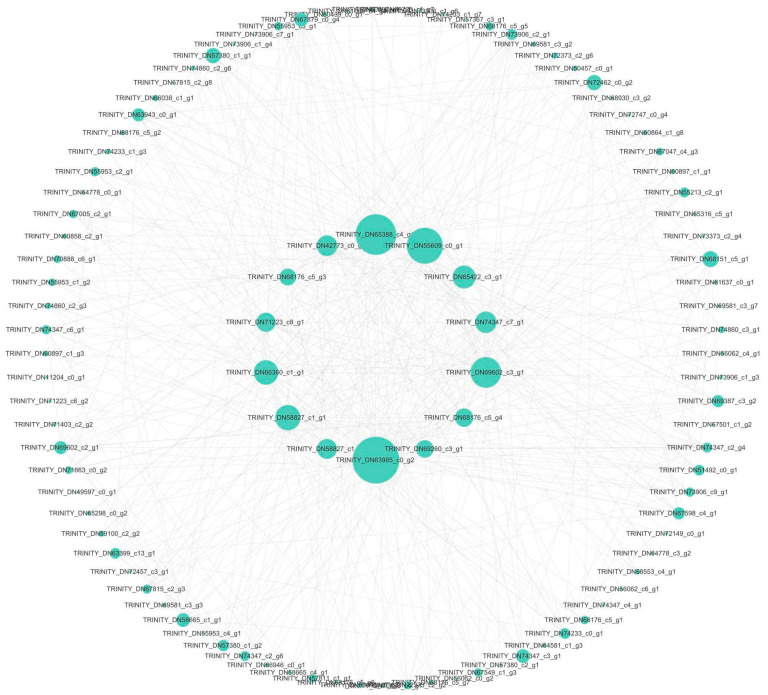
Co−expression network diagram of genes in Yellow module.

## Data Availability

The datasets for this study can be found in the Appendix A. The transcriptome raw data of *P. suffruticosa* ‘Hu Hong’ data were uploaded on the NCBI SRA under the following accession numbers: PRJNA1079236.

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
