# Peer review of "Mining Heat-Resistant Key Genes of Peony Based on Weighted Gene Co-Expression Network Analysis"

_genes, 2024, doi:10.3390/genes15030383_

Round 1

Reviewer 1 Report

Comments and Suggestions for Authors

In the paper “Mining Heat-Resistant Key Genes of Peony Based on Weighted  Gene Co-Expression Network Analysis”, several transcriptome datasets were primarily used to identify highly co-expressed clusters of genes (modules) in peony leaves under high temperature conditions, by WGCNA.

Due to the unclear writing style, the authors are requested to provide clarification on the following points:.

1.- In the introduction section (38-42, 54-56), the authors imply that the transcriptome sequencing data analyzed in this study were derived from previous findings. Additionally, throughout the manuscript, various references are cited to suggest the potential origin of the transcriptome dataset used (Cao, 2012; Zhang, 2019; Yao, 2022, etc.). Unfortunately, that references provided by the authors are not available to the public.

However, in the materials and methods section, the authors outline the acquisition of the transcriptome dataset without citing any references, creating an impression that the dataset is an integral part of this current paper.

Therefore, could the authors provide explicit references to their work or offer a description detailing the acquisition of the transcriptome data?

2.-  In addition to the ambiguity surrounding the transcriptome data source, the authors assert that all datasets supporting the conclusions of this article are included within the article (453, 454). However, no links to any datasets were found. Given that published data should be easily verifiable by any research group, it is customary to deposit databases in a public repository, such as NCBI.

WGCNA allowed the authors to generate several hypotheses. Yet, should the transcriptome dataset pertain to another published paper, it is strongly suggested to the authors to complete the work with a validation in independent data or in designed validation experiments. The two potential core genes related to high temperature stress are natural candidates for further validation.

Author Response

Dear Reviewer1:

Thank you for your letter and for the reviewers’ comments concerning our manuscript entitled “Mining Heat-Resistant Key Genes of Peony Based on Weighted Gene Co-Expression Network Analysis”. Those comments are all valuable and very helpful for revising and improving our paper, as well as the important guiding significance to our researches. We have studied comments carefully and have made correction which we hope meet with approval. Revised portion are marked in yellow in the paper. 

Reviewer 2 Report

Comments and Suggestions for Authors

The paper has serious problems, both formal and in the research design. Specifically:

- According to methods, authors have data from several points of heat stress. "n 0 h, 2 h, 6 h, 12 h, 24 h group ( blank control CK, heat stress treatment HS ) sampling 3 times, a total of 30 samples" but then data are presented Heat vs control. So The results are the same at different times? how are the three samples chosen? From which times? This is a major problem for the credibility of the results.

- The clustering of the results is difficult to understand, and the notation based in colors, confusing. Is not clear which is the distinctive trait for each cluster. The level of significance? the correlation? The function of the genes? The explanation is very poor. 

Figure 1 and figure 2: lettering is too small. Legend too concise. What does any color mean in the heat map, which are the axis legends? What is the dendogram?

English must be corrected. many sentences difficult to understand. i.e. line 98-99 or the first sentence of the abstract, which seems and unfinished sentence.

Lines 30-33: You are describing severeal medicinal effects, but citation number 5 is an horticultural journal. Please cite a proper medical journal where the effect has been checked or do not mention it. 

Comments on the Quality of English Language

English must be thoroughly revised

Author Response

Dear  Reviewer 2:

Thank you for your letter and for the reviewers’ comments concerning our manuscript entitled “Mining Heat-Resistant Key Genes of Peony Based on Weighted Gene Co-Expression Network Analysis”. Those comments are all valuable and very helpful for revising and improving our paper, as well as the important guiding significance to our researches. We have studied comments carefully and have made correction which we hope meet with approval. Revised portion are marked in yellow in the paper. 

Reviewer 3 Report

Comments and Suggestions for Authors

This manuscript investigates the genetic basis of heat tolerance in Paeonia ostii 'Huhong' through RNA-Seq analysis under high-temperature stress using weighted co-expression network analysis (WGCNA), key modules and genes associated with heat response. The study holds merit as it investigates heat-tolerant genes in economically important peony plant, which is relevant for scaling up production and developing resilient varieties to withstand environmental challenges. However, the manuscript in its current form needs many modifications before it could be accepted for publication. I therefore recommend reconsider after major revision. Below is my detailed review

1.      Introduction:

The introduction lacks clarity at several points, with repeated phrases hindering readers' comprehension. Terms like "temperature poisoning" are unclear and should be clarified. Instead of using "high temperature," the authors should specify the heat tolerance capacity of Peony.

2.      Materials and Methods:

The authors have not sufficiently detailed the methods used to obtain transcriptome data. If the analysis relies on data generated by the authors, they should elaborate on the growing conditions, the process of generating raw data, and the subsequent steps involved in data processing. Alternatively, if the data is sourced from previous literature, the authors should explicitly provide details regarding the raw data. The origin of the data remains ambiguous in this context.

At many places, the text is somewhat unclear due to sentence framing and use of technical terminology without explanations. There are a few grammatical errors and awkward phrasing that could be improved for better readability and comprehension.

Line 98-111: Please explain why authors choose to use Arabidopsis as a reference species to compare the annotations results of protein database in peony. Also, the text mentions setting the protein interaction detection basis to 0.95 but doesn't clarify how this value was determined or its significance. Similarly, the mention of eliminating proteins with "low correlation connectivity" lacks explanation or context.

The material and methods section does not provide details on how transcription factor analysis was conducted.

3.      Results:

Please provide more detailed information in the legends of the figures to enhance their clarity and comprehensiveness.

Line 128-129: What are CK and HS the authors are talking about?

Line139-140: Please provide details regarding the 48 enrichment terms mentioned by the authors. Additionally, clarify whether the "modules" referred to by the authors are indeed GO categories?

Line153: what are the enriched 46 pathways? name them and how they relate to the heat tolerance study.

Line 313- 318: Please provide references when talking about a gene function.

The authors have used informal language instead of scientifically precise terminology in several instances. For example, "more regular" in line 180, and "highly homologous" in line 211

The blue, brown, and yellow clusters are not clearly defined and clearly explained.

4.      Discussion:

The discussion section presents several issues that could be improved for clarity and coherence. Some information is repeated unnecessarily, such as the examples of previous studies on WRKY53 and HsfB2b. Streamlining these repetitions would improve readability. Also, there is a need to integrate the study's findings with previous research and expanding on future research directions.

Comments on the Quality of English Language

English language needs improvement throughout the manuscript

Author Response

Dear Reviewer 3:

Thank you for your letter and for the reviewers’ comments concerning our manuscript entitled “Mining Heat-Resistant Key Genes of Peony Based on Weighted Gene Co-Expression Network Analysis”. Those comments are all valuable and very helpful for revising and improving our paper, as well as the important guiding significance to our researches. We have studied comments carefully and have made correction which we hope meet with approval. Revised portion are marked in yellow in the paper. 

Round 2

Reviewer 1 Report

Comments and Suggestions for Authors

The data analysis presented in this work is good, the results are intriguing, and the authors have demonstrated an enhancement in the description of the results and methods section. It is now evident that the transcriptome dataset is an integral component of this paper. However, the manuscript still exhibits a confusing writing style, and in certain paragraphs, it sounds wordy. It is strongly recommended that the authors effectively organize their ideas and make it clear and concise before publication.

Example:

Regarding phrases that suggest a second research project:

-In the early stage of our research group, the transcriptome of peony leaves under high temperature stress was studied, and a large number of differentially expressed genes were obtained and enriched [7]. However, due to the lack of complete genomic information, there are too many differential genes screened out. In order to solve this problem, this study used the WGCNA method to modularly integrate them (Lines 35-40).

-Based on the previous transcriptome sequencing results, this study used the…  (line 51).

I suggest removing any words that don’t add value and instead confuse the reader:

-The transcriptome of peony leaves under high temperature stress was studied for our research group, a large number of differentially expressed genes were obtained and enriched [7]. However, due to the lack of complete genomic information, there are too many differential genes screened out. In order to solve this problem, we used the WGCNA method to modularly integrate them….

-Based on the transcriptome sequencing results, we used the….

Great writing is a major factor in whether you receive a favorable or unfavorable grade, and in whether you gain academic credibility or lose it.

Author Response

(The authors gave the same response as above.)

Reviewer 2 Report

Comments and Suggestions for Authors

I appreciate the changes made by the authors, but the main problems of the manuscript are still there. Sampling at different times but presenting the results as a whole lacks of biological relevance as the response is time dependent.

Also the color grouping is quite confusing, I suggest to include a table explaining what dies each color mean,

Most of the figures are impossible to read as lettering is too small-

Comments on the Quality of English Language

English needs a revision.

Author Response

Dear Reviewer2:

Thank you for your letter and for the reviewers’ comments concerning our manuscript entitled “Mining Heat-Resistant Key Genes of Peony Based on Weighted Gene Co-Expression Network Analysis”. Those comments are all valuable and very helpful for revising and improving our paper, as well as the important guiding significance to our researches. We have studied comments carefully and have made correction which we hope meet with approval. Revised portion are marked in yellow in the paper.

Round 3

Reviewer 2 Report

Comments and Suggestions for Authors

Paper has been greatly improved, now it can be published

Comments on the Quality of English Language

English still needs some revision